# Determination of Adequate Substrate Water Content for Mass Production of a High Value-Added Medicinal Plant, *Crepidiastrum denticulatum* (Houtt.) Pak & Kawano

**Song-Yi Park [1,2,†], Jongyun Kim [3,†] and Myung-Min Oh [1,2,*]**

1  Division of Animal, Horticultural and Food Sciences, Chungbuk National University, Cheongju 28644, Korea; 1songyi1@gmail.com
2  Brain Korea 21 Center for Bio-Resource Development, Chungbuk National University, Cheongju 28644, Korea
3  Division of Biotechnology, Korea University, Seoul 02841, Korea; jongkim@korea.ac.kr
*  Correspondence: moh@cbnu.ac.kr; Tel.: +82-43-261-2530
†  These authors contributed equally to this work.

**Abstract:** The effects of substrate water content on the growth and content of bioactive compounds in *Crepidiastrum denticulatum* were evaluated. Three-week-old seedlings were subjected to four levels of substrate water content (20%, 30%, 45% and 60%) and maintained for 5 weeks. Growth parameters at 5 weeks of transplanting were significantly higher with the 45% substrate water content treatment than with the other treatments. In addition, photosynthetic rate, stomatal conductance and transpiration rate increased significantly and the highest sap flow rate during the day was observed in 45% substrate water content. Total phenolic content and antioxidant capacity per shoot increased significantly with substrate water content, increasing from 20% to 45% and decreased again at 60%. Antioxidant capacity and total hydroxycinnamic acids (HCAs) content per unit dry weight of plants under the 60% treatment were significantly higher than those under the 45% treatment; however, their content per shoot was the highest under the 45% treatment. Thus, 45% substrate water content is a suitable condition for the growth of *C. denticulatum* and had positive effects on phenolic content, antioxidant capacity, and HCAs content. These results could be useful for the mass production of high-quality *C. denticulatum* in greenhouses or plant factories capable of controlling the water content of the root zone.

**Keywords:** antioxidant capacity; bioactive compounds; growth; hydroxycinnamic acids; hydroponics

---

## 1. Introduction

Water is one of the crucial factors for plant growth and development accounting for 80%–90% and over 50% of the fresh weight of herbaceous and woody plants, respectively [1]. Temporary water deficit in plants causes turgor loss and stomatal closure that inhibits basic metabolic processes, including photosynthesis. Excessive or constant water deficit generates reactive oxygen species (ROS) in plants, causing oxidative stress and photoinhibitory damage that eventually result in necrosis and programmed cell death [2–4]. In general, plants adapt to certain levels of water stress by promoting the biosynthesis of secondary metabolites with antioxidant properties. Water availability around the root zone directly affects the physiological and biochemical responses of plants. Therefore, controlling water content of the substrate is an important cultural practice that directly influences crop yield and quality in horticultural plant-production.

*Crepidiastrum denticulatum* H. is an annual or biennial species (family: Compositae) and grows naturally in East Asia and South Korea. It contains a large amount of bioactive compounds including various hydroxycinnamic acids (HCAs) such as chlorogenic acid, 3,5-di-O-caffeoylquinic acid (3,5-DCQA), chicoric acid and caftaric acid. Several previous studies have reported that *C. denticulatum* extracts have high anti-oxidative, anti-fatty liver and anti-obesity properties [5–8], and health functional foods for improving liver function have also been produced using the extract of *C. denticulatum*. One of the ecological characteristics of *C. denticulatum* is its sensitivity to the water condition around the root zone with soft rot diseases in the leaf often occurring under conditions of excessive water with frequent rainfall during summer. The occurrence of soft rot diseases was observed more often in wet soil compared to deeper and well-drained soil, resulting in the poor growth and quality of *C. denticulatum* [9].

The appropriate limitation of water supply to the root zone can be used as a cultivation technique to produce high-quality crops by promoting the biosynthesis of bioactive compounds without the inhibition of growth. In previous studies, temporary mild water stress did not inhibit the growth of lettuce, water dropwort and tomato, with a simultaneous increase in the content of polyphenolic compounds such as chicoric acid (in lettuce), anthocyanin (in water dropwort) and hydroxycinnamic acid and flavonoids (in tomato), which have antioxidant properties [10–12]. In addition, the biosynthesis of secondary metabolites in medicinal plants was also promoted by water deficit stress. For example, the content of hyperforin, the major bioactive compound in St. John's wort, was increased by 200% at 12 days of water stress compared to that in the control [13]. Water deficit treatment with a 50% reduction in irrigation increased the silymarin content of milk thistle by 170% compared to that in the control [4,13]. Recently, greenhouses or plant factories capable of controlling the root environment using various water-related sensors have been established, and the demand for high value-added crops such as medicinal plants is increasing. However, little research has been reported on the proper conditions for such cultivation. In this respect, research on the favorable water content of the substrate is needed for the stable mass production of high-quality *C. denticulatum* as a raw material for pharmaceutical products or functional foods.

Therefore, the objective of this study was to evaluate the growth and content of bioactive compounds in *C. denticulatum* according to different water content levels of the substrate. Through this study, we determined the favorable water content level of the substrate for the stable mass production of high-quality *C. denticulatum*.

## 2. Materials and Methods

### 2.1. Plant Materials and Experimental Conditions

*C. denticulatum* seeds collected from Pyeongchang in Korea were sown according to the method described in Park et al. [14]. Seedlings were grown for 3 weeks in a growth chamber with the following conditions: air temperature, 20 °C; relative humidity, 60%; white LEDs, PPFD 200 µmol m$^{-2}$ s$^{-1}$; and light period; 16 h. Seedlings were transferred to a greenhouse for acclimation 3 days before transplanting. A total of 48 seedlings with 2–3 true leaves were transplanted into individual square plastic pots (10 × 10 × 11 cm; L × W × H) filled with commercial horticultural substrate (Myung-Moon, Dongbu Hannong Co., Seoul, Korea). The average air temperature and relative humidity of the greenhouse were 19.7 ± 0.1 °C; and 48.5 ± 0.5% (± S.E.), respectively, and average daily light integral was 9.1 ± 1.6 mol m$^{-2}$ d$^{-1}$ during the entire experimental period.

### 2.2. Treatments of Substrate Water Content

Four substrate water content levels of 20, 30, 45 and 60% were used and each level was maintained for 5 weeks starting 1 week after transplanting. Twenty-four soil water sensors (EC-5, METER group, Pullman, WA, USA) were inserted individually in 24 pots (six plants per treatment), and real-time data of volumetric water content (v/v) were collected via a data logger (CR 1000, Campbell Scientific, Logan, UT, USA) connected to the sensors. To measure the volumetric water content of the substrate, the soil

water sensors were calibrated using a formula obtained from a preliminary study. The relay driver (SDM-16AC/DC, Campbell Scientific, Logan, UT, USA) connected with the data logger opened the solenoid valves of the irrigation line to supply the nutrient solution (nutrient solution for *C. denticulatum*, EC 2.0 dS m$^{-1}$, pH 5.5) [14] to the pots when the volumetric water content measured by the soil water sensor was lower than the set value. Two drip pins connected with a pressure compensated emitter (2L/H, Netafim, Tel Aviv, Israel) were inserted into both sides of a pot to supply the nutrient solution evenly. Figure 1 shows the changes in the volumetric water content of the substrates with the different treatments during the entire experimental period.

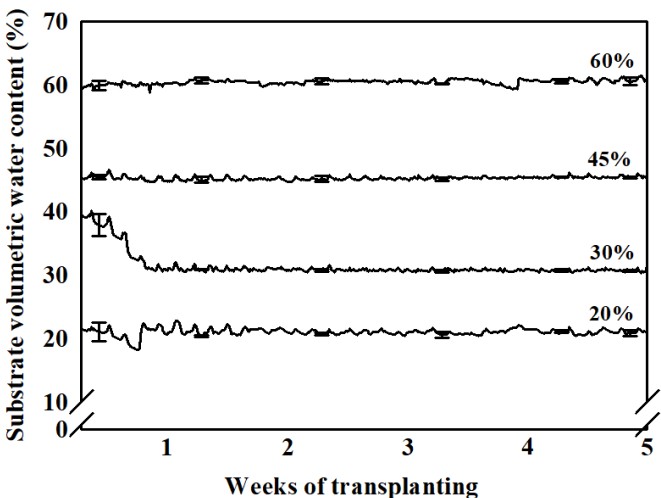

**Figure 1.** Average volumetric water content of the substrate at 20%, 30%, 45% and 60% water content treatments for 5 weeks. Lines and bars indicate the means and standard errors, respectively (*n* = 6).

### 2.3. Plant Growth Parameters

Plant growth parameters were investigated at 5 weeks after transplanting. The shoot and root were separated at the basal end and the substrate of the roots was removed by washing under running water. The remaining water was blotted using paper towels. Fresh weights of the shoot and root were measured using an electronic scale (Si-234, Denver Instrument, Bohemia, NY, USA). Shoot dry weight was measured after freeze-drying at −75 °C; for over 72 h using a lyophilizer (Alpha 24 LSCplus, CHRIST, Osterode am Harz, Germany) and root dry weight was measured after hot-air drying at 70 °C; for over 72 h. Leaf length and leaf width of the largest leaf of the plants were measured using a ruler. The leaf shape index was calculated as leaf length/leaf width. The total leaf area was measured using a leaf area meter (LI-2050A, Li-Cor, Lincoln, NE, USA).

### 2.4. Photosynthetic Parameters

Photosynthetic rate, stomatal conductance and transpiration rate of *C. denticulatum* were measured using a portable photosynthesis system (LI-6400, LI-COR, Lincoln, NE, USA) for 2 h, starting at 10 a.m. (3 h after sunrise) 4 weeks after transplanting. The leaf chamber conditions were set at 24 °C; block temperature, 500 μmol mol$^{-1}$ reference $CO_2$, 400 μmol s$^{-1}$ air flow and 308 μmol m$^{-2}$ s$^{-1}$ PPFD (average PPFD in the morning during the experiment). Six plants with fully expanded leaves per treatment were measured.

Shoots were freeze-dried after harvest and used for chlorophyll content analysis. Shoots were pulverized using a grinder (Tube Mill control, IKA, Wilmington, NC, USA). A sample of powder (40 mg) and 4 mL acetone (80%, v/v) was mixed, and then the mixture was sonicated for 15 min. The supernatant obtained by centrifugation at 15,000 × g for 2 min was diluted four times with acetone (80%, v/v). The absorbance of the solution was measured with a spectrophotometer (UV-1800, Shimadzu, Kyoto, Japan) at 663.6, 646.6 and 750 nm and chlorophyll a, chlorophyll b and chlorophyll a + b were calculated using the following equation [15].

$$\text{Chlorophyll a} = 12.25\text{Absorbance}^{(663.6-750)} - 2.55\text{A}^{(646.6-750)} \tag{1}$$

$$\text{Chlorophyll b} = 20.31\text{A}^{(646.6-750)} - 4.91\text{A}^{(663.6-750)} \tag{2}$$

$$\text{Chlorophyll a} + \text{b} = 17.76\text{A}^{(646.6-750)} + 7.34\text{A}^{(663.6-750)} \tag{3}$$

### 2.5. Sap Flow

The sap flow rate was measured to determine the effect of substrate water content on water absorption and transpiration. A micro sap flow sensor (MSF_UM, Telofarm, Seoul, Korea) was inserted into the stem of a fully expanded leaf at 4 weeks after treatment and data were continuously collected for 5 days. Sap flow values were recorded every 2 min by the data logger.

### 2.6. Total Phenolic Content and Antioxidant Capacity

To investigate the effects of various levels of substrate water content on the biosynthesis of secondary metabolites in *C. denticulatum*, samples were collected immediately after harvest to analyze total phenolic content and antioxidant capacity. Total phenolic content and antioxidant capacity were analyzed using a powdered sample (40 mg) obtained by grinding the freeze-dried whole shoot. Total phenols were extracted, and antioxidant capacity analyzed, as previously described in Park et al. [14]. Total phenolic content was expressed as the content of gallic acid (mg) either per unit dry weight or per shoot. Antioxidant capacity was expressed as trolox (6-Hydroxy-2,5,7,8-tetramethylchromane-2-carboxyl acid) (mM) either per unit dry weight or per shoot.

### 2.7. Hydroxycinnamic Acids

Caftaric acid, chlorogenic acid, caffeic acid, chicoric acid and 3,5-DCQA were extracted from a freeze-dried powder sample (100 mg) using an ultrasonicator (SK5210HP, Young Jin Corporation, Gunpo, Korea) with 70% aqueous ethanol for 90 min. Individual hydroxycinnamic acids were analyzed using a high-performance liquid chromatograph 185 (YL9100, Young Lin Instrument Co., Ltd., Anyang, Korea) according to the method previously described in Park et al. [14]. Standard curves were obtained using caftaric acid (ChemFaces, Hubei, China), chlorogenic acid, caffeic acid, 3,5-DCQA (Sigma-Aldrich, St. Louis, MO) and chicoric acid (Avention, Incheon, Korea) and the content of each compound was expressed as mg per unit dry weight of the shoot.

### 2.8. Statistical Analysis

In this experiment, we used a randomized complete block design with three blocks and four plants were randomly arranged in each block for each treatment. Twelve plants per treatment were used for the analysis of growth parameters, chlorophyll content, total phenolic content, antioxidant capacity and HCAs content. Photosynthetic rate, stomatal conductance and transpiration rate were measured using six plants per treatment. Statistical analysis of the results was conducted using Statistical Analysis System, 9.2 Version, SAS Institute, Cary, NC, USA (SAS). Analysis of variance (ANOVA) and Tukey's Studentized Range Test (HSD) were used to determine the statistical significance among treatments.

## 3. Results

### 3.1. Plant Growth Parameters

Different levels of substrate water content affected the growth of the shoot and root of *C. denticulatum* significantly (Figure 2; Figure 3). As substrate water content increased from 20% to 45%, fresh and dry weights of the shoot and root increased significantly, but decreased at 60%. In particular, the shoot and root biomass were higher by 2.4 and 1.8 times, respectively, with the 45% substrate water content treatment than with the 20% treatment which showed the lowest growth performance. Changes in leaf length and leaf area also showed a similar pattern to that observed in shoot growth; the highest value

was recorded with the 45% substrate water content treatment. In the case of leaf width, significantly higher values were observed with the 30% and 45% substrate water content treatments than with the 20% treatment. Leaf shape index, an indicator of leaf shape, was also influenced by substrate water content and had the lowest value in plants treated with the 20% of substrate water content.

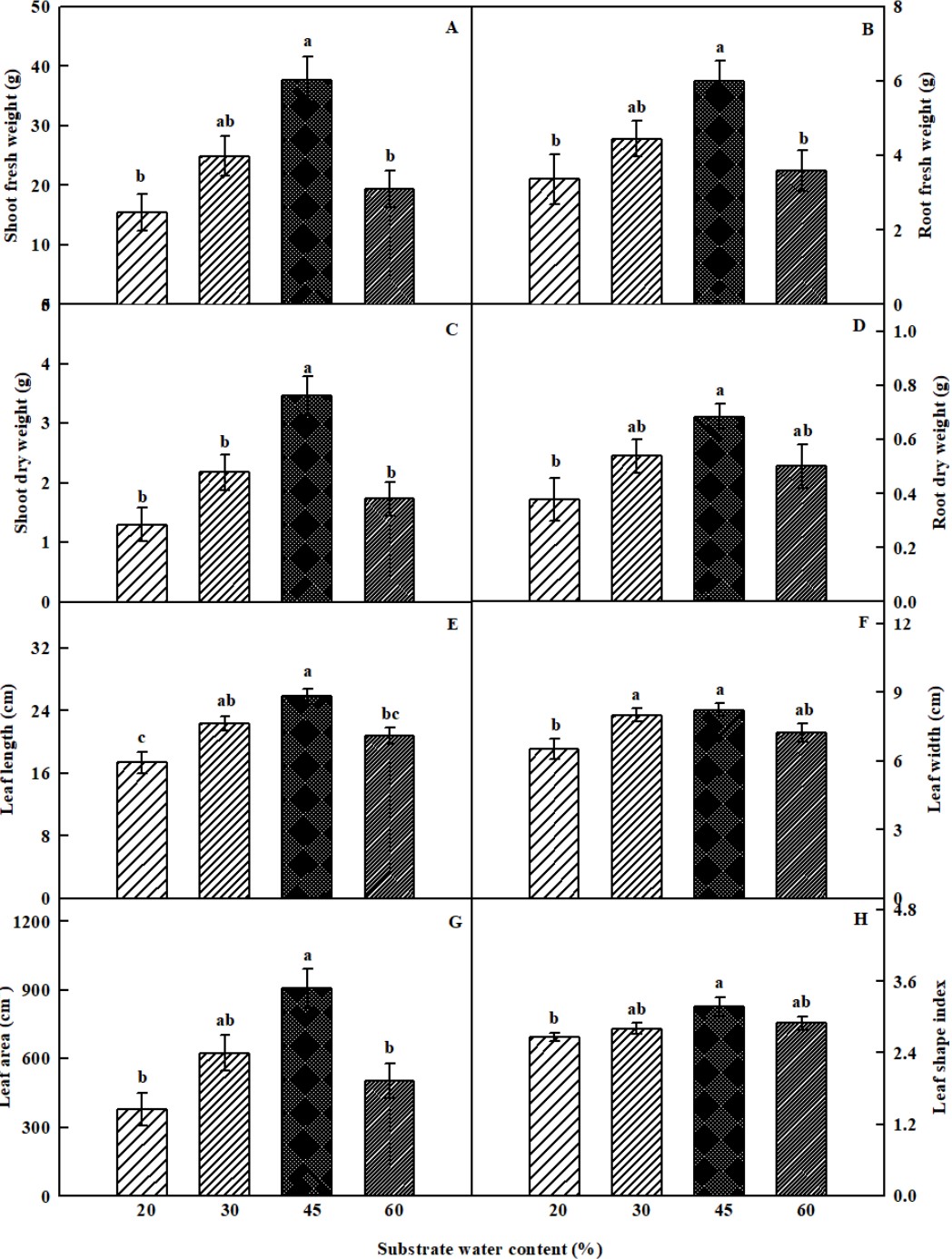

**Figure 2.** Effect of substrate water content on growth parameters: Fresh and dry weighs of shoot (**A** and **C**) and root (**B** and **D**), leaf length (**E**), leaf width (**F**), leaf area (**G**) and leaf shape index (**H**) of under four different substrate water content levels for 5 weeks. The data indicate the means ± S.E. ($n = 12$). Different letters above the bars indicate statistical difference by Tukey's Studentized Range Test at $p < 0.05$.

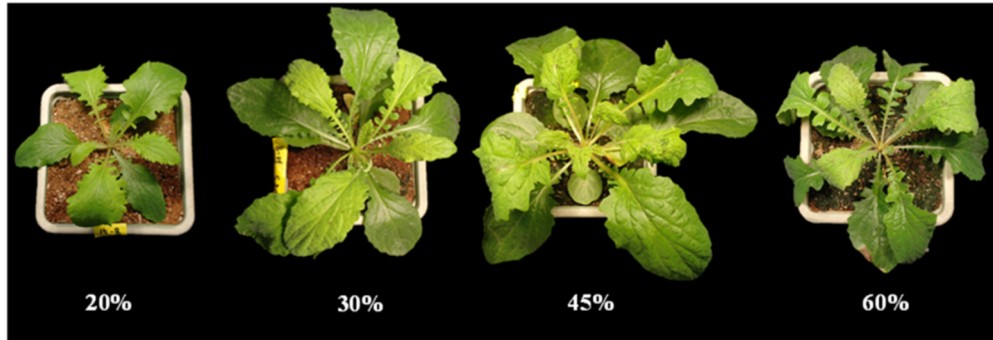

**Figure 3.** *Crepidiastrum denticulatum* grown under different substrate water content levels at 5 weeks after transplanting.

### 3.2. Photosynthetic Parameters

Photosynthetic parameters that support the increase in biomass were measured at 4 weeks after transplanting in *C. denticulatum* grown with different substrate water content (Figure 4A–C). The photosynthetic rate of plants grown under the 45% substrate water content treatment, which exhibited excellent shoot biomass, was higher than that of plants grown under the other treatments, and the photosynthetic rate of plants under the 60% substrate water content treatment was not different from that of plants under the 20% and 30% treatments. As substrate water content increased from 20% to 45%, stomatal conductance gradually increased, and then decreased again at 60%. The change in transpiration rate was similar to that observed for stomatal conductance and was significantly higher in plants grown under the 30% and 45% substrate water content treatments than in plants under the 20% and 60% treatments. The chlorophyll content of *C. denticulatum* per unit dry weight was not affected by the water content of the substrate (Figure 4D).

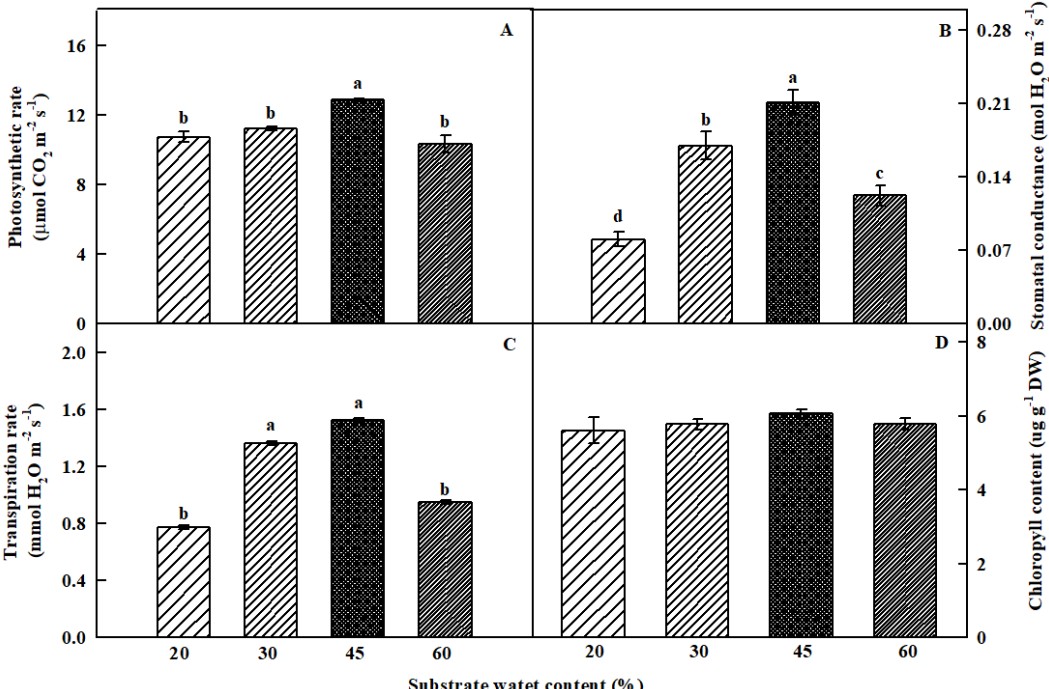

**Figure 4.** Effect of substrate water content on photosynthetic parameters: Photosynthetic rate (**A**), stomatal conductance (**B**), transpiration rate (**C**) and chlorophyll content (**D**) of *Crepidiastrum denticulatum* grown under four different substrate water content levels at 4 weeks after transplanting. The data indicate the means ± S.E. (photosynthetic parameters; n = 6 and chlorophyll content; n = 12). Different letters above the bars indicate statistical difference by Tukey's Studentized Range Test at $p < 0.05$.

### 3.3. Sap Flow

After sunrise, at around 8 a.m., sap flow values of plants in all treatments began to increase rapidly, and all values slowly increased with repeated increases and decreases until around 2 p.m. (Figure 5). All sap flow values gradually declined from after 3 p.m. until sunset. Similar to the results observed for the photosynthetic parameters, the sap flow value in plants grown under the 45% substrate water content treatment was the highest during the day. In particular, in plants grown under the 45% treatment, the sap flow value measured from 3 p.m. until sunset was clearly distinguished from that in plants grown under other treatments and remained high.

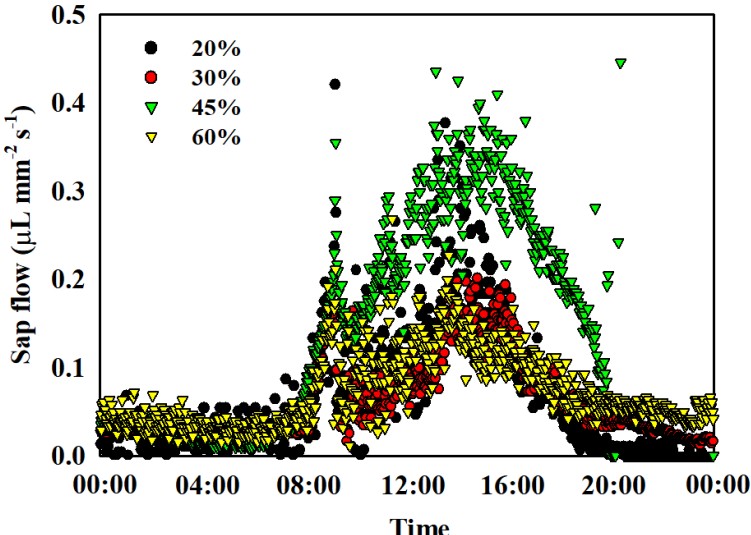

**Figure 5.** Sap flow values of *Crepidiastrum denticulatum* grown under four different substrate water content levels at 4 weeks after transplanting. A sap flow sensor was inserted into the stem of one plant per treatment. Sap flow values were recorded every 2 min by the data logger.

### 3.4. Total Phenolic Content and Antioxidant Capacity

The four levels of substrate water content showed no significant difference in total phenolic content per unit dry weight. However, the antioxidant capacity of plants under the 60% treatment was higher than that of plants under the 45% treatment and there was no difference in plants under the 20% to 45% treatments (Figure 6A,B). Total phenolic content and antioxidant capacity per shoot increased significantly from 20% to 45% of substrate water content treatments and decreased again under the 60% treatment, similar to the results observed for the shoot growth. In particular, the highest phenolic content and antioxidant capacity were also found in plants under the 45% treatment, which had the best shoot growth.

### 3.5. Hydroxycinnamic Acids

After harvest, four types of hydroxycinnamic acids (HCAs) were analyzed (Figure 7). Total HCAs content per unit dry weight was significantly higher in plants grown under the 60% substrate water content treatment, whereas chicoric acid content was significantly the lowest in plants grown with the 45% treatment. There was no difference between plants grown under the other three treatments. Individual HCAs and total HCAs content per shoot were significantly the highest in plants grown under the 45% substrate water content treatment that had the superior shoot biomass. In particular, the total HCAs content of plants grown under the 45% treatment was approximately three times higher than that of plants grown under the 20% treatment.

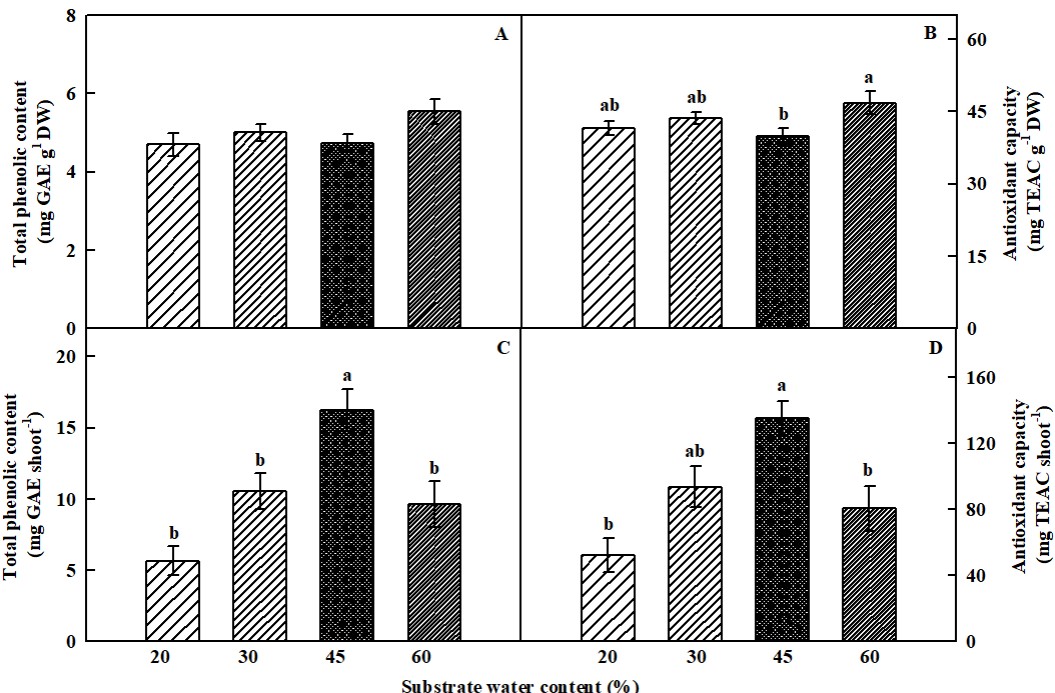

**Figure 6.** Effect of substrate water content on phenolic content and antioxidant capacity: Total phenolic content and antioxidant capacity per unit dry weight (**A** and **C**) and per shoot (**B** and **D**) of *Crepidiastrum denticulatum* grown under four different substrate water content levels for 5 weeks. The data indicate the means ± S.E. ($n$ = 12). Different letters above the bars indicate statistical difference by Tukey's Studentized Range Test at $p < 0.05$.

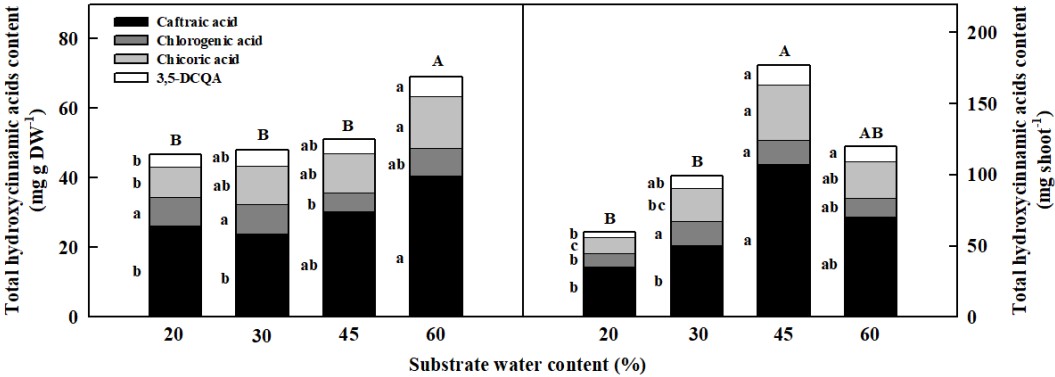

**Figure 7.** Effect of substrate water content on hydroxycinnamic acids' (HCAs) content: Total HCAs' content per unit dry weight (**A**) and per shoot (**B**) of Crepidiastrum denticulatum grown under four different substrate water content levels for 5 weeks. The data indicate the means ± S.E. ($n$ = 12). Different lowercase letters indicate statistical difference in each individual compound by Tukey's Studentized Range Test at $p < 0.05$. Different uppercase letters indicate statistical difference in total HCAs by Tukey's Studentized Range Test at $p < 0.05$.

## 4. Discussion

In the early stage of soil drought and flooding stresses, root signals limit the movement of water and nutrients to the growing zones; these changes result in the collapse of the water potential gradient between the xylem and growing cells [16,17]. A continuous drought condition eventually causes dehydration of the shoot and stimulates the biosynthesis of abscisic acid (ABA) around the root meristem. ABA is transported to the shoot via the xylem stream and causes stomatal closure, thereby preventing water loss from leaves. This results in a restriction in the transport of large amounts

of water, minerals and various chemical compounds from the roots to the shoots. Drought stress increases the content of ABA in leaves, not only by promoting the biosynthesis of ABA but also by accelerating the ABA catabolic pathway [18–20]. In contrast, excessive water content in soil or the substrate causes an oxygen-poor environment (hypoxia), which becomes a major factor in inhibiting root respiration. This inhibits metabolic activities and ATP production in plants. These plant responses limit the supply of energy for the growth of the root and eventually result in poor plant growth [2,19]. In addition, flooding stress induces the accumulation of toxic compounds such as ethanol, lactic acid and acetaldehyde, and these compounds not only suppress plant growth but also cause adverse effects such as root dysfunction and low soil redox potential [21,22]. In this experiment, the inhibition of shoot and root growth under the 20% and 60% substrate water content conditions could be explained as being caused by drought and flooding stresses, respectively (Figure 2). These results imply that the substrate water content between these values, 45%, in which plants showed the most effective growth, may be adequate for the growth of *C. denticulatum*.

Stomatal closing due to drought stress as described above prevents water loss in the leaves, while it hinders transpiration and the allocation of photosynthetic assimilates, resulting in a reduction in photosynthetic rate [21,23]. Hypoxia conditions in soil caused by flooding restrict water absorption by roots and root hydraulic conductance, leading to stomatal closure [24,25]. In addition, Ahsan et al. [3] reported that soil flooding increases photorespiration and/or decreases the activities of RuBP and RuBP activase, which is one of the main reasons for the reduction in photosynthetic rate. In our study, substrate water content of 20% (drought) and 60% (flooding) decreased stomatal conductance and transpiration rate in the leaves of *C. denticulatum* (Figure 4B,C). The observed sap flow values also supported the results of stomatal conductance and transpiration rate. During the day, plants under the 20% and 60% substrate water content treatments had lower sap flow values than plants under the 45% treatment (Figure 5). In this experiment, the decrease in transpiration rate owing to soil drought and flooding seemed to have a direct effect on photosynthetic inhibition (Figure 4A). Mutava et al. [26] reported that drought-tolerant genotypes accumulate more ABA than drought-susceptible genotypes, leading to an increased stomatal closure in dry soil. In this experiment, the stomatal conductance of plants under the 20% treatment was significantly lower than that of plants under the 60% treatment (Figure 4B), suggesting that *C. denticulatum,* which naturally grows at the foot of a mountain in dry conditions, may be more sensitive to excessive soil water content than to dry soil. In this study, chlorophyll content was not significantly affected by substrate water content (Figure 4D). In general, constant or excessive water stress decreases chlorophyll content and ultimately accelerates leaf senescence [3]. However, it should be considered that the water level and duration conditions used in this study were not severe enough to affect a change in chlorophyll content. Plants have complex defensive mechanisms to survive in tough external environments, among which the antioxidant system plays an important role in overcoming environmental stresses. Soil-water-related stresses promote the generation of reactive oxygen species (ROS) owing to the limitation of photosynthetic processes, which can activate the antioxidant system [27–29]. The typical antioxidant bioactive compounds of *C. denticulatum* are types of HCAs including caftaric acid, chicoric acid, chlorogenic acid and 3,5-DCQA [5–7,14]. The results of this experiment showed that antioxidant capacity and total HCAs per dry weight of plants under the soil flooding treatment (60%) were higher than those of plants under other treatments (Figures 6B and 7A). This result can also be explained by the concentration effect because of growth inhibition. However, considering that shoot dry weights of 20% and 30% substrate water content treatments were similar to 60% treatment, not only the inhibition of photosynthesis by flooding stress but also hypoxia-produced ROS and toxic substances are thought to activate the biosynthesis pathway of secondary metabolites [21]. At 20% water content of the substrate, total phenolic content and HCAs content did not increase, probably owing to a lack of photosynthetic assimilates. However, although antioxidant capacity and total HCAs content per dry weight of plants grown under the 60% substrate water content treatment were significantly higher than in plants under the 45% treatment, their content per shoot were the highest under the 45% substrate water content treatment because of a remarkable

increase in the shoot dry biomass. Another study of the cultivation of *C. denticulatum* used the capillary wick culture system described in a previous study [14,30]. The results also showed that shoot fresh weight, photosynthetic rate, total phenolic content and antioxidant capacity per shoot of plants grown under 45% substrate water content were significantly the highest among treatments (Figure S1). These results showed that shoot biomass of *C. denticulatum* is directly related to the content of bioactive compounds, and that 45% water content of the substrate can increase not only the growth, but also the antioxidant capacity and phenolic content.

In conclusion, we confirmed that the growth and bioactive compounds of *C. denticulatum,* which is used as a plant-derived raw material for functional food, can be influenced by the water content of the substrate. The water content of 45% in the substrate increased the biomass of the shoot and root and increased phenolic content, antioxidant capacity and HCAs content per shoot. The possibility of using hydroponics for native plants was also verified by another experiment using the capillary wick culture system. The results of the current study are expected to be useful for the stable mass production of high-quality *C. denticulatum* in greenhouses or plant factories capable of controlling water content in the root zone.

**Supplementary Materials:** The following are available online at http://www.mdpi.com/2073-4395/10/3/388/s1, Figure S1: effects of capillary wick culture system.

**Author Contributions:** Methodology, investigation, resources, formal analysis, data curation, software, writing—original draft preparation, visualization, S.-Y.P.; methodology, conceptualization, formal analysis, software, writing—original draft preparation, J.K.; supervision, funding acquisition, validation, writing—review and editing, M.-M.O. All authors have read and agreed to the published version of the manuscript.

**Funding:** This work was supported by Korea Institute of Planning and Evaluation for Technology in Food, Agriculture, Forestry and Fisheries (IPET) through the Agriculture, Food and Rural Affairs Research Center Support Program, funded by the Ministry of Agriculture, Food and Rural Affairs (MAFRA) (717001-07-02-HD240).

**Conflicts of Interest:** The authors declare no conflict of interest. The funders had no role in the design of the study; in the collection, analyses, or interpretation of data; in the writing of the manuscript, or in the decision to publish the results.

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
