# Peer review of "Determination of Adequate Substrate Water Content for Mass Production of a High Value-Added Medicinal Plant, Crepidiastrum denticulatum (Houtt.) Pak & Kawano"

_agronomy, doi:10.3390/agronomy10030388_

Round 1

Reviewer 1 Report

In general, the article is well written.

There is only one suggestion about the length of sentence. In this article, there are many long sentences that exceeded 20 words. For example, line 211-215 (more than 50 words in one sentence). It would be better to separate it to 2 or 3 short sentences for easier reading and understanding. 

Line 209-211, better to move this description to `Materials and Methods`.

In discussion part:

It would be clearer for readers if the author inserts a subtitle/subheading for each paragraph.

Line 245, `The generated ABA`?

Line 280, long-term what? This sentence is not clear.

Line 300-304, it is quite misleading. The author should describe the system of literature `30` and your additional experiment separately.  

Please also describe how your additional experiment was conducted in detail in the supplemental materials.

Author Response

  1. There is only one suggestion about the length of sentence. In this article, there are many long sentences that exceeded 20 words. For example, line 211-215 (more than 50 words in one sentence). It would be better to separate it to 2 or 3 short sentences for easier reading and understanding.

- Line 40-41, 215-217, 218-219, 283, 288, 299-304: We revised as you suggested.

  1. Line 209-211, better to move this description to `Materials and Methods`.

- Line 138-140: Lines were moved to `Materials and Methods`.

In discussion part:

  1. It would be clearer for readers if the author inserts a subtitle/subheading for each paragraph.

- We think that it is not necessary to insert a subtitle to separate paragraphs because there are some discussions leading from paragraph to the next paragraph.

  1. Line 245 ‘The generated ABA`?

- Line 252: It refers to the biosynthesis ABA mentioned in the previous sentence. To avoid confusion, ‘generated’ was deleted.

  1. Line 280, long-term what? This sentence is not clear.

- Line 287: We changed that word (constant).

  1. Line 300-304, it is quite misleading. The author should describe the system of literature `30` and your additional experiment separately.

- Line 310-315: We revised that sentence. Another study of the cultivation of C. denticulatum using the capillary wick culture system previously described in Park et al. [14]. The results also showed that shoot fresh weight, photosynthetic rate, total phenolic content and antioxidant capacity per shoot of plants grown under 45% substrate water content were significantly the highest among treatments (Figure S1).

  1. Please also describe how your additional experiment was conducted in detail in the supplemental materials.

- We revised that sentence. Figure S1. Effects of capillary wick culture system: Three-week-old seedlings were transplanted to pots with three different water contents of substrate (20, 45, 50%) and nutrient solution was subirrigated using capillary wicks inserted into the bottom of pots for 6 weeks. Shoot fresh weight (A), photosynthetic rate (B), total phenolic content (C) and antioxidant capacity (D) of Crepidiastrum denticulatum grown under 3 different substrate water content levels using the capillary wick culture system for 6 weeks. The data indicate the means ± S.E. (n = 4).

Reviewer 2 Report

Suggestions:

1) Introduction,

line 39: antioxidant properties (delete the s of antioxidants)
line 42: replace "horticultural fields" with "horticultural plant-production"

2) Materials and Methods

line 80: please specify the growing substrate
line 102: maybe Plant-Growth-Parameters is the better wording than Plant Growth Charactistics (latter refers to plant growth analysis more)

3) Figure 3 and 3.2. Potosynthetic Parameters:
line 180 and the text of figure: delete the "at" of "at 5 weeks" and "at 4 weeks".

4) Maybe a bit more information on C. denticulatum and its medical value - this species is rarely known and maybe not in use in Europe (and other regions in the world?)

5) Please give the use of the term "hydroponics" some critical reflections - yes, it is used in different ways in the horticultural world. At least here it usually refers to production systems without the use of solid growing substrates (hydroponics, aeroponics).

6) "Plant factories" are not new in horticulture but turned out to be of interest in the recent years. Some comments on their potential might create additional value.

Thanks for your submission - this is a solid and excellent contribution and work.

Author Response

Introduction,

  1. line 39: antioxidant properties (delete the s of antioxidants)

- Line 39: We deleted the s of antioxidants.

  1. line 42: replace "horticultural fields" with "horticultural plant-production"

- Line 42: We replace "horticultural fields" with "horticultural plant-production".

Materials and Methods

  1. line 80: please specify the growing substrate

- Line 81: We revise the word (commercial horticultural substrate). The composition was not given because commercial horticultural substrate (mixed soil: cocopeat, peatmoss, vermiculite, perlite, zeolite, fertilizer) is commonly used in the word.

  1. line 102: maybe Plant-Growth-Parameters is the better wording than Plant Growth Charactistics (latter refers to plant growth analysis more)

- Line 16-17, 104, 105, 159, 166: We replace ‘Charactistics’ with ‘parameters’ in the manuscript.

Figure 3 and 3.2. Potosynthetic Parameters:

  1. line 180 and the text of figure: delete the "at" of "at 5 weeks" and "at 4 weeks".

- We think it is better not to revise it because it is expressly fine.

  1. Maybe a bit more information on denticulatum and its medical value - this species is rarely known and maybe not in use in Europe (and other regions in the world?)

- We have already introduced C. denticulatum in our previous paper and pharmacological effects have also been also demonstrated in reference papers. We filled in a little more information about this plant in introduction.

  1. Please give the use of the term "hydroponics" some critical reflections - yes, it is used in different ways in the horticultural world. At least here it usually refers to production systems without the use of solid growing substrates (hydroponics, aeroponics).

- Capillary wick culture is one of the hydroponic systems. We used mixed substrate (perlite:peatmoss=7:3, v/v). Two capillary wicks (1.5 ×0.14 cm [W × T], Non-woven fabrics) were inserted into the bottom for each pot and dipped into nutrient solution. Nutrient solution rises into the growing substrate through the capillary wicks described in previous study. In hydroponic system, many types of mediums such as rockwool, oasis cube, cocopeat, ferlite, vermiculite, and peat moss can be used.  

  1. "Plant factories" are not new in horticulture but turned out to be of interest in the recent years. Some comments on their potential might create additional value.

- This manuscript is focused on the effect of water control on the growth and bioactive compounds of a medicinal plant. Basically I agree with your opinion in that this technique directly can be applied to plant factory industry. However, we described the potential effect on plant factories in introduction and conclusion. We thought that that is enough.   
